

# Clinical impact of heterogeneously distributed tumor-infiltrating lymphocytes on the prognosis of colorectal cancer

Lu Liu[1], Meiling Long[2], Shengyuan Su[1], Lijun Wang[1] and Jintao Liu[3]

[1] Shenzhen People's Hospital of Baoan District, Shenzhen, China
[2] Xinyi People's Hospital, Maoming, China
[3] Shenzhen Baoan Traditional Chinese Medicine Hospital Group, Shenzhen, China

## ABSTRACT

**Background.** Tumor-infiltrating lymphocytes (TILs) exist in various malignancies, and have been viewed as a promising biomarker to predict the efficacy and outcome of treatment. However, the marked inter- and intra-tumor heterogeneity of TILs has resulted in some confusion regarding their impact on the prognosis of colorectal cancer (CRC).

**Methods.** In this study, 78 CRC patients were enrolled and the CD3+ and CD8+ TILs densities at the tumor center (TC), the invasive margin (IM) and the tumor stroma (TS) were assessed by immunohistochemical staining. Their associations with clinicopathological features and progression free survival (PFS) were analyzed to evaluate the predictive and prognostic values of TILs.

**Results.** TILs were mainly distributed along the invasive margin. High density of TILs in tumor center and invasive margin was associated with smaller tumor size (CD3+TILs$^{IM}$), reduced tumor invasion (CD3+TILs$^{IM}$), absence of lymph node metastasis (CD3+TILs$^{IM}$ and CD8+TILs$^{TC}$), earlier stage (CD3+TILs$^{IM}$ and CD8+TILs$^{IM}$), and lower tumor grade (CD3+TILs$^{IM}$ and CD8+TILs$^{TC}$). However, stromal TILs were not associated with any clinicopathological features. Kaplan–Meier survival analysis revealed that high densities of TILs always correlated with prolonged patient survival. The pathological N stage, CD3+ TILs$^{IM}$ and CD8+ TILs$^{TC}$ were found to be independent prognostic indicators. Additionally, early-stage CRC patients who developed recurrence after surgery, showed a higher CD3+/CD8+ TILs ratio in invasive margin. In the present study, it was clarified that CD3+ and CD8+ TILs were heterogeneously distributed in tumor tissues of CRC. The increase in intratumoral and peritumoral TILs had been shown to be strongly predictive of improved clinical outcome. More importantly, the immune signatures enabled to stratify early-stage CRC patients with high risk of recurrence, highlighting the prognostic power of TILs.

# INTRODUCTION

Colorectal cancer (CRC) is one of the most prevalent cancers worldwide, which accounts for approximately 10% of cancer-related mortality (*Siegel et al., 2022*; *Brenner, Kloor & Pox, 2014*). Currently, advances in cancer treatment involving tumor surgery, radiotherapy,

Corresponding author
Jintao Liu, jtliu118@163.com

chemotherapy and immunotherapy, have considerably improved the overall survival of CRC patients. However, there is a significant difference in clinical outcome among patients with the same pathophysiological evaluation and treatment decisions, demonstrating the limitation of current system of pathophysiological evaluation for CRC (*Nagtegaal, Quirke & Schmoll, 2011*; *Galon et al., 2013*; *Galon et al., 2014*). Notably, it has been found that approximately 10–30% of CRC patients with stage II disease developed local recurrence after surgical resection (*Nozawa et al., 2018*; *Glaire et al., 2019*). Therefore, more effective prognostic strategies are urgently needed to improve outcome prediction and treatment selection.

Tumor cells live in a complex milieu of cellular components comprising endothelial cells, stromal cells and tumor-infiltrating lymphocytes (TILs), which strongly influence tumor development and progression (*Hanahan & Weinberg, 2011*; *Fridman et al., 2017*; *Hinshaw & Shevde, 2019*). Notably, increasing evidence has suggested that these infiltrated immune cells are very diverse from patient to patient (*Galon et al., 2013*; *Galon et al., 2014*). Therefore, the analysis of the location, density as well as functional state of the different immune cell populations in tumors may be a promising source of effective diagnostic and prognostic biomarkers (*Galon et al., 2006*). For CRC, numerous studies have showed the prognostic value of TILs (*Galon et al., 2006*; *Pagès et al., 2009*; *Mlecnik et al., 2011*; *Laghi et al., 2009*; *Frey et al., 2010*; *Zhou et al., 2019*). These experiences demonstrated that the presence of abundant T cells, especially cytotoxic CD8+ T cells, in tumor tissues is typically viewed as an indicator of good prognosis (*Pagès et al., 2009*). However, the marked intra-tumor and inter-tumor heterogeneity of TILs has posed significant challenges to the development of predictive strategies incorporating TILs features (*Fridman et al., 2012*). Currently, relatively little information is available on the impact of TILs distribution pattern on the prognosis of CRC. In this study, the number of both CD3+ and CD8+ TILs in the tumor center (TC), the tumor stroma (TS) and the invasive margin (IM), were counted separately, and therefore a more accurate evaluation of their relationship with clinical characteristics and outcome may be achieved, which helps for guiding personalized treatment.

## MATERIALS & METHODS

### Patients

This study was approved by the ethics committee of People's Hospital of Baoan District (Date: March 8, 2021; NO: BYL20210337). The informed consent for research was waived since the following three conditions can be met: (1) it is difficult to obtain informed consent, (2) the research is of high social importance, (3) the research does not infringe the rights and interests of study subjects.

Seventy-eight CRC patients were retrospectively collected from the pathology archive of People's Hospital of Baoan District between 2017 and 2019. All enrolled patients underwent complete surgical resection so that their surgical margins were microscopically negative for residual tumors. None of the patients had been treated with preoperative chemotherapy or radiotherapy. Patients who were death or recurrence within 1 month after surgery,

were excluded. Their clinical and histopathological data were reviewed by two experienced gastrointestinal pathologists according to the American Joint Committee on Cancer the tumor-node-metastasis (TNM) staging system. The observation time in this study was the interval between diagnosis and last contact (death, local recurrence or lost to follow-up).

## Immunohistochemistry

Formalin-fixed paraffin-embedding tissue blocks were serially sectioned at 5-$\mu$m thickness. Prior to immunohistochemistry, tissue slides were assessed by hematoxylin and eosin staining to confirm the presence of tumor and adjacent cells. Then, they were deparaffinized in xylene, ethanol and water, and were treated with citrate buffer (0.01 M, pH 6.0) and heated in a microwave oven at 90 °C for 30 min. After blocking by 5% (v/v) human serum, tissue slides were incubated with the rabbit polyclonal anti-CD3 antibody (ab5690; dilution, 1:100; Abcam, Cambridge, UK) or anti-CD8 antibody (ab4055; dilution, 1:200; Abcam) at 37 °C for 120 min, followed by incubation with the HRP-conjugated secondary antibody(goat anti-rabbit IgG H&L; ab214880; Abcam) for 15 min at 37 °C .

## Cell quantification

3D Histech Midi Scanner System was used to digitalize stained slides that were viewed in the Panoramic Viewer system. Five visual fields with the most abundant infiltration of T cells were selected from each slice to calculate the mean density of TILs. Numbers of positively immunostained cells were double-blindly evaluated by two experienced pathologists.

## Statistical analysis

The patients' parameters that were taken into account for statistical analysis contained clinical (serum level of carcinoembryonic antigen (CEA) and carbohydrate antigen 19-9 (CA19-9), chemotherapy regimen, and recurrence as well as age, sex), histopathologic (tumor size, histological type, pathologic TNM, and differentiation), and immunologic variables (CD3+ and CD8+ cells). Pearson's $\chi 2$ test or Fisher's exact (two-sided) test was used for examining the difference between groups. Survival curves were estimated by the Kaplan–Meier method, and the comparison between groups were performed using the log-rank test. Multivariate logistic regressions were used to identify clinicopathological features associated with the recurrence of patients with early stage diseases. Proportional hazards models (Cox) were performed to identify the variables associated with patient's survival. All statistical analysis described above was conducted using SPSS 22.0, and $p$-value <0.05 was thought to be statistically significant.

# RESULTS

## Study population

We retrospectively analyzed 78 CRC patients without preoperative treatment. Their clinicopathological characteristics were presented in Table 1. Histologically, most patients were diagnosed as adenocarcinoma (59/78, 75.6%), and presented with deep tumor invasion (T3-T4, 67/78, 85.9%). The incidence of lymph node metastasis was 42.3% (33/78). Primary tumors were commonly located on the left side (56/78, 71.8%). 38

patients (48.7%) experienced local recurrence after surgery. The median duration of PFS was 31.1 months (19.46–35.49 months, 95% confidence interval [CI]). Among the clinicopathological parameters, the depth of tumor invasion (T stage, $P = 0.033$), lymph node invasion (N stage, $P < 0.0001$) and TNM stage ($P < 0.0001$) were significantly associated with PFS (Table 1).

## TILs distribution pattern in CRC

In the present study, to characterize the distribution pattern of TILs, the whole slide was digitally imaged and segmented into three regions for counting the stained TILs. We found both CD3+ and CD8+ TILs displayed a similar distribution pattern. They were mainly located along the invasive margin (CD3+TILs$^{IM}$, 1,063.22 ± 99.13; CD8+TILs$^{IM}$, 386.75 ± 102.01), and, to a lesser extent, at the tumor center (CD3+TILs$^{TC}$, 122.54 ± 48.18; CD8+TILs$^{TC}$, 70.82 ± 27.03) and at the tumor stroma (CD3+TILs$^{TS}$, 336.61 ± 100.80; CD8+TILs$^{TS}$, 165.41 ± 37.22). Statistical significances were observed among the density of TILs at these difference areas (Figs. 1G and 1H), indicating that TILs were unevenly distributed in CRC. Although intratumoal TILs showed marginal increases in CRC patients with higher peritumoral TILs, these did not reach statistical significance (CD3+TILs$^{TC}$ vs CD3+TILs$^{IM}$, $P = 0.535$; CD3+TILs$^{TC}$ vs CD3+TILs$^{TS}$, $P = 0.240$; CD8+TILs$^{TC}$ vs CD8+TILs$^{IM}$, $P = 0.093$; CD8+TILs$^{TC}$ vs CD8+TILs$^{TS}$, $P = 0.058$). The ratio of CD3+ to CD8+ TILs was also evaluated. Tumor center displayed a significantly lower CD3+/CD8+ ratio compared with that in the invasive margin ($P < 0.001$, Fig. 1I), suggesting that the composition of infiltrated T cells was significantly different between tumor center and invasive margin.

## The correlation between TILs densities and clinicopathological features

The cohort was divided into low and high-density groups according to the median threshold, and spearman's rank correlation was used to explore the correlation between TILs densities and clinicopathologic features. Only intratumoral and peritumoural TILs were considered in the following analysis since there was no significant correlation between TILs in tumor stroma and clinicopathological features (Table S1). As shown in Table 2, the associations of clinicopathological features with CD3+ and CD8+ TILs at the same region were not identical. For CD3+TILs, low density of CD3+ TILs$^{IM}$ was associated with larger tumor size (odds ratio [OR], 0.309; 95% CI [0.121–0.786]), deeper tumor invasion (OR, 0.181; 95% CI [0.036–0.897]), positive lymph node metastasis (OR, 0.303; 95% CI [0.118–0.778]) and advanced stage (OR, 0.245; 95% CI [0.095–0.634]). Intratumoral CD3+, CD3+ TILs$^{TC}$, was more frequently observed in primary tumor located at the right side (OR, 0.324; 95% CI [0.109–0.964]). For CD8+ TILs, intratumoral CD8+, CD8+ TILs$^{TC}$, was significantly higher in tumors with the well differentiation (OR, 0.228; 95% CI [0.066–0.782]) and without lymph node metastasis (OR, 0.347; 95% CI [0.138–0.871]). Peritumoural CD8+, CD8+TILs$^{IM}$, was also inversely correlated with tumor stage (OR, 0.326; 95% CI [0.126–0.843]). No significant association was observed between densities of TILs with age, sex, as well as the level of serum CEA and CA19-9.

**Table 1** Patients clinicopathological features and their correlations with PFS.

|  |  | PFS (95% CI) | *p*-value |
|---|---|---|---|
| **Age** |  |  | 0.923 |
| <50 | 21 | 36.5(28.9–47.1) |  |
| >50 | 57 | 41.5(32.9–43.9) |  |
| **Sex** |  |  | 0.421 |
| Male | 41 | 36.4(29.3–43.4) |  |
| Female | 37 | 40.5(34.4–46.6) |  |
| **CEA(ng/ml)** |  |  | 0.586 |
| Negative (<5) | 48 | 38.9(32.8–45.1) |  |
| Positive (>5) | 30 | 36.4(28.4–44.3) |  |
| **CA19-9 (U/ml)** |  |  | 0.319 |
| Negative (<37) | 69 | 39.4(34.2–44.6) |  |
| Positive (>37) | 9 | 33.4(22.6–44.1) |  |
| **Tumor location** |  |  | 0.188 |
| Left sided | 56 | 36.2(30.5–41.8) |  |
| Right sided | 22 | 42.6 (34.5–50.8) |  |
| **Tumor size (cm)** |  |  | 0.885 |
| <5 | 43 | 39.2(32.9–45.5) |  |
| >5 | 35 | 38.1(30.1–46.0) |  |
| **Histological type** |  |  | 0.262 |
| adenocarcinoma | 59 | 40.3(35.2–45.4) |  |
| mucinous adenocarcinoma | 15 | 33.7(21.2–46.3) |  |
| signet ring cell carcinoma | 4 | 25.3(4.5–64.1) |  |
| **T stage** |  |  | **0.033** |
| T1+T2 | 11 | 50.5(34.6–66.4) |  |
| T3+T4 | 67 | 36.3(31.6–41.1) |  |
| **N stage** |  |  | **<0.0001** |
| N0 | 45 | 47.8(41.9–53.7) |  |
| N1+N2 | 33 | 29.3(18.7–35.3) |  |
| **TNM Stage** |  |  | **<0.0001** |
| I+II | 43 | 50.7(45.5–55.9) |  |
| III+ IV | 35 | 28.2(22.5–33.9) |  |
| **Histological grade** |  |  | 0.171 |
| Well | 21 | 36.9(19.5–54.3) |  |
| Moderately | 40 | 44.3(39.3–49.3) |  |
| Poorly | 17 | 23.8(14.2–33.4) |  |

Notes.
Bold indicated a statistically significant.
PFS, progression free survival; CI, confidence interval; CEA, carcinoembryonic antigen; CA19-9, carbohydrate antigen 19-9.

## The correlation with total patient outcome

Table 3 showed the results of univariate survival analysis for the clinicopathological features. We found that the degree of tumor invasion (HR, 0.325; 95% CI [0.158–0.669]; $p = 0.002$), and lymph node metastasis (HR, 0.260; 95% CI [0.137–0.495], $p = 0.0001$)

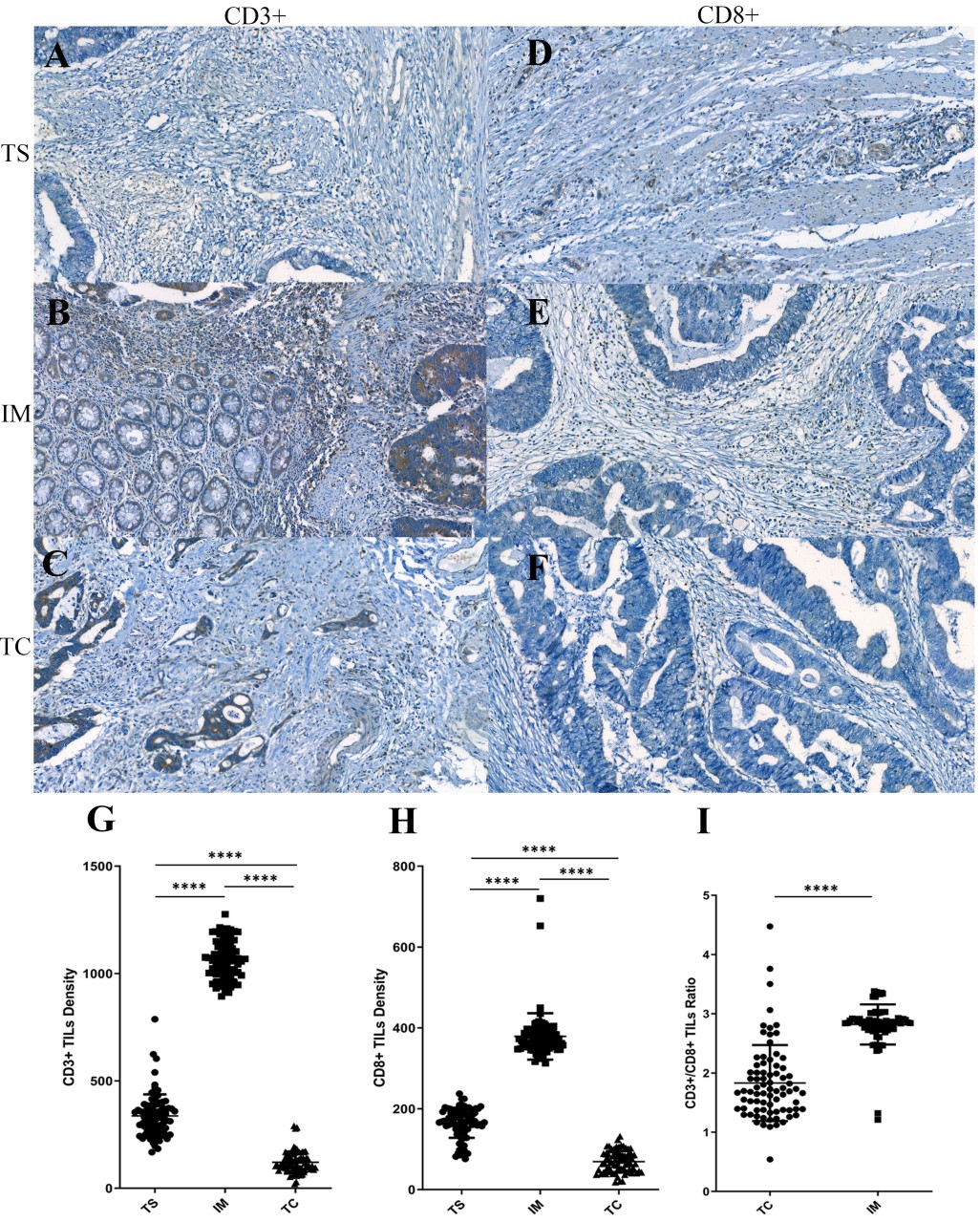

**Figure 1 The distribution pattern of tumor-infiltrating lymphocytes (TILs) in colorectal cancer (CRC).** (A–F) Representative immunohistochemistry stainings for CD3 (left) and CD8 (right) in the tumor stroma (TS), in the invasive margin (IM) and in the tumor center (TC). (G–H) Densities of CD3+ and CD8+ TILs in three different regions. (I) CD3+/CD8+ TILs ratio in TC and IM.

were the factors associated with CRC patient's survival. Kaplan–Meier analysis showed that the higher densities of CD3+ and CD8+ TILs at the tumor center and invasive margin were implicated in favorable outcomes (Fig. 2).

**Table 2  Correlations between TILs and clinicopathological features.**

| | CD3+ | | | | CD8+ | | | |
| | TC | | IM | | TC | | IM | |
| | OR (95% CI) | p-value | OR (95% CI) | p-value | OR (95% CI) | p-value | OR (95% CI) | p-value |
|---|---|---|---|---|---|---|---|---|
| **Age** | | | | | | | | |
| <50 vs. >50 | 0.675 (0.246–1.851) | 0.549 | 0.877 (0.322–2.389) | 0.991 | 0.516 (0.185–1.436) | 0.671 | 0.391 (0.137–1.113) | 0.235 |
| **Sex** | | | | | | | | |
| Male vs. Female | 0.902 (0.371–2.195) | 0.803 | 0.734 (0.301–1.790) | 0.109 | 1.362 (0.558–3.322) | 0.148 | 1.228 (0.504–2.988) | 0.795 |
| **CEA** | | | | | | | | |
| Negative vs. Positive | 0.805 (0.322–2.007) | 0.293 | 0.647 (0.258–1.621) | 0.123 | 1.242 (0.498–3.098) | 0.830 | 0.518 (0.205–1.309) | 0.097 |
| **CA19-9** | | | | | | | | |
| Negative vs. Positive | 0.458 (0.106–1.981) | 0.82 | 1.286 (0.318–5.202) | 0.952 | 0.247 (0.047–1.275) | 0.236 | 0.712 (0.192–3.141) | 0.578 |
| **Tumor location** | | | | | | | | |
| Left vs. Right | 0.324 (0.109–0.964) | **0.024** | 0.966 (0.341–2.740) | 0.850 | 0.600 (0.221–1.629) | 0.071 | 0.776 (0.288–2.086) | 0.294 |
| **Tumor size (cm)** | | | | | | | | |
| <5 vs. >5 | 1.11 (0.454–2.708) | 0.679 | 0.309 (0.121–0.786) | **0.008** | 0.901 (0.369–2.201) | 0.977 | 0.732 (0.299–1.792) | 0.151 |
| **Histological type** | | | | | | | | |
| Adenocarcinoma vs. others | 0.364 (0.121–1.087) | 0.145 | 0.870 (0.309–2.449) | 0.593 | 0.492 (0.169–1.425) | 0.376 | 0.574 (0.195–1.686) | 0.441 |
| **T stage** | | | | | | | | |
| T1+T2 vs. T3+T4 | 0.808 (0.224–2.908) | 0.933 | 0.181 (0.036-0897) | **0.004** | 0.522 (0.139–1.953) | 0.874 | 0.333 (0.081–1.365) | 0.063 |
| **N stage** | | | | | | | | |
| N0 vs. N1+N2 | 0.729 (0.296–1.795) | 0.313 | 0.303 (0.118–0.778) | **0.001** | 0.347 (0.138–0.871) | **0.027** | 0.589 (0.238–1.459) | 0.076 |
| **TNM Stage** | | | | | | | | |
| I+II vs. III+ IV | 0.593 (0.241–1.461) | 0.384 | 0.245 (0.095–0.634) | **0.0002** | 0.732 (0.299–1.792) | 0.419 | 0.326 (0.126–0.843) | **0.006** |
| **Histological grade** | | | | | | | | |
| Well/Moderately vs. Poorly | 0.330 (0.104–1.055) | 0.112 | 0.148 (0.038–0.572) | **0.012** | 0.228 (0.066–0.782) | **0.019** | 0.462 (0.151–1.411) | 0.261 |

**Notes.**
Bold indicated a statistically significant.
TC, tumor center; IM, invasive margin; OR, odds ratio; CI, confidence interval; CEA, carcinoembryonic antigen; CA19-9, carbohydrate antigen 19-9.

Multivariate logistic regression was performed to identify independent prognostic factors, in which all features found to be prognostic in univariate analysis were included. Multivariate analysis revealed that N stage (HR, 0.474; 95% CI [0.260–0.862]; $P = 0.017$), CD3+ TILs$^{IM}$, (HR, 1.893; 95% CI [1.071–3.460]; $P = 0.028$) and CD8+ TILs$^{TC}$ (HR, 2.050; 95% CI [1.211–3.704]; $P = 0.013$) showed independent prognostic significance.

**Table 3  Univariate and multivariate analysis for the factors associated with progression free survival.**

| | Univariate analysis | | Multivariate analysis | |
|---|---|---|---|---|
| | HR (95% CI) | *p*-value | HR (95% CI) | *p*-value |
| **T stage** | | | | |
| T1+T2 *vs.* T3+T4 | 0.325 (0.158–0.669) | **0.002** | 0.588 (0.322–1.064) | 0.085 |
| **N stage** | | | | |
| N0 *vs.* N1+N2 | 0.260 (0.137–0.495) | **0.0001** | 0.474 (0.260–0.862) | **0.017** |
| **CD3+ TILs$^{TC}$** | | | | |
| Low *vs.* High | 2.243 (1.362–4.589) | **0.004** | 1.471 (0.870–2.479) | 0.152 |
| **CD3+ TILs$^{IM}$** | | | | |
| Low *vs.* High | 2.468 (1.340–4.546) | **0.003** | 1.893 (1.071–3.460) | **0.028** |
| **CD8+ TILs$^{TC}$** | | | | |
| Low *vs.* High | 2.842 (1.518–5.320) | **0.001** | 2.050 (1.211–3.704) | **0.013** |
| **CD8+ TILs$^{IM}$** | | | | |
| Low *vs.* High | 1.992 (1.207–4.043) | **0.013** | 1.376 (0.682–3.436) | 0.323 |

**Notes.**
Bold indicated a statistically significant.
HR, hazard ratio; CI, confidence interval.

These observations also confirmed that high densities of CD3+ TILs$^{IM}$ and CD8+ TILs$^{TC}$, correlated with good outcome.

## The correlation with early-stage patient outcome

Of the 43 patients with stage I/II CRC, 7 patients (16.28%) developed disease recurrence. No significant difference was observed between recurrence and non-recurrence patients in their clinicopathological features, including age, sex, serum CEA and CA19-9 level, tumor size and the depth of tumor invasion. We next studied whether the characteristic of TILs could offer a prognostic indicator for the recurrence of early-stage CRC patients. As shown in Table 4, recurrent CRC patients had a higher CD3+/CD8+ TILs ratio in invasive margin, whereas no statistical difference was seen for densities of CD3+ or CD8+ TILs in different regions. These results also suggested that compared with the absolute number or density of TILs, the composition of TILs had a more important impact on the prognosis of early-stage CRC patients.

## DISCUSSION

The presence of abundant TILs within the tumor microenvironment is generally indicative of the activation of antitumor immune response (*Fridman et al., 2012*). Importantly, mounting evidence indicates that the composition and distribution pattern of TILs are able to change with the dynamic process of cancer immunoediting which determines

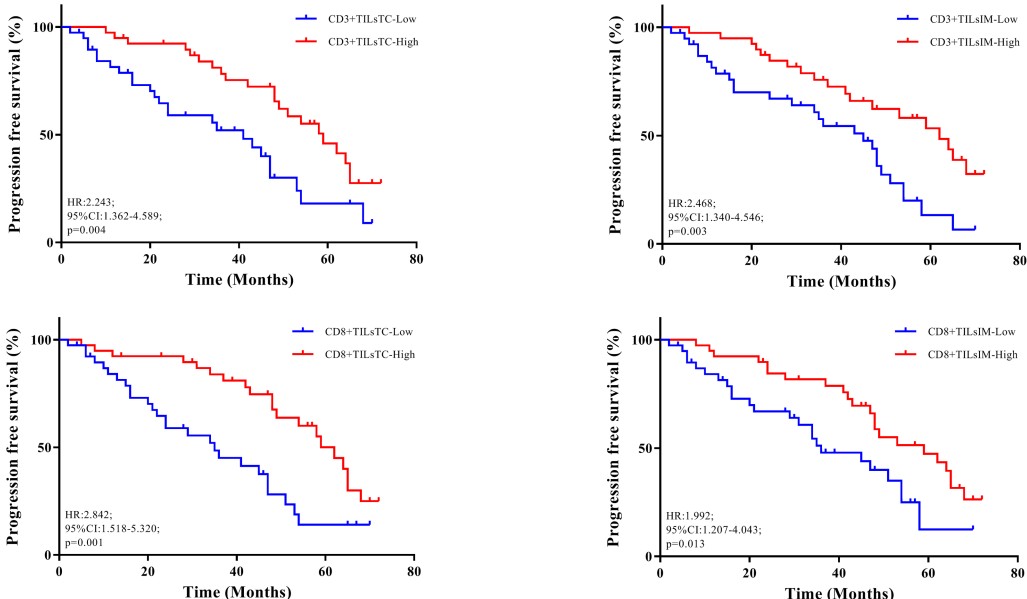

**Figure 2** **Relationship between progression free survival and density of tumor-infiltrating lymphocytes (TILs).** (A–B) Kaplan–Meier curves illustrated the duration of progression free survival according to the density of CD3+ TILs in tumor center (TC) and invasive margin (IM), respectively. (C–D) Kaplan–Meier curves illustrated the duration of progression free survival according to the density of CD8+ TILs in TC and IM, respectively.

the progression and outcome of cancer (*Hanahan & Weinberg, 2011*; *Fridman et al., 2017*). Therefore, the characteristic of TILs has been proposed as an effective predictor of outcome in cancer patients and can help for guiding therapeutic intervention. In the present study, we performed a detailed immunohistochemical evaluation of TILs in three areas, including the tumor core, the invasive margin and the tumor stroma. We found that the higher densities of CD3+TILs$^{IM}$ and CD8+TIL$^{TC}$ correlated with earlier stage, reduced incidence of lymph node metastasis, the absence of tumor invasion, and thus, lower risk of progression. These results suggested that more abundance infiltration of tumor tissue by T cells had significant antitumor benefits and improved patient survival, which was in agreement with previous reports (*Galon et al., 2006*; *Pagès et al., 2009*; *Mlecnik et al., 2011*; *Laghi et al., 2009*; *Frey et al., 2010*; *Zhou et al., 2019*).

In the present study, TILs were detected in every tumor tissue of CRC patient, regardless of their TNM stages, demonstrating that TILs participated in all stages of tumor progression. We found that TILs dominantly localized to the invasive margin rather than the tumor center, indicating that these cells influenced tumor development through indirect action of secreted soluble mediators, such as cytokines, chemokines and other bioactive factors, rather than through direct T cell-mediated cytotoxicity (*Nagarsheth, Wicha & Zou, 2017*). Additionally, we observed a decrease in the ratio of CD3+/CD8+ TILs in tumor center compared with that in invasive margin, suggesting that intratumoral TILs, TILs$^{TC}$ were comprised of both CD8+ and CD4+ cells, whereas CD4+ cells predominated over CD8+ cells in invasive margin. Taken together, these observations indicated that the spatial

**Table 4   Multivariate analysis of risk factors for recurrence of CRC patients with early stage diseases.**

| | Multivariate analysis | |
| --- | --- | --- |
| | HR (95% CI) | *p*-value |
| **CD3+ TILs$^{TC}$** | | |
| Low *vs.* High | 0.877 (0.574–1.333) | 0.529 |
| **CD3+ TILs$^{IM}$** | | |
| Low *vs.* High | 0.662 (0.284–1.538) | 0.416 |
| **CD8+ TILs$^{TC}$** | | |
| Low *vs.* High | 0.775 (0.709–2.083) | 0.250 |
| **CD8+ TILs$^{IM}$** | | |
| Low *vs.* High | 0.526 (0.226–1.219) | 0.283 |
| **CD3+/CD8+ TILs$^{TC}$** | | |
| Low *vs.* High | 1.739 (0.709–4.255) | 0.354 |
| **CD3+/CD8+ TILs$^{IM}$** | | |
| Low *vs.* High | 1.694 (1.031–2.703) | **0.027** |

**Notes.**
Bold indicated a statistically significant.
HR, hazard ratio; CI, confidence interval.

distribution and composition of TILs were highly heterogeneous, which might explain the variable impact of TILs on prognosis in CRC. For example, *Chiba et al. (2004)* pointed out the number of intratumoral CD8+ T cells had a more significant impact on patients' survival than that of CD8+ T cells in other locations within the tumor. *Menon et al. (2004)* found that CD8+ TILs along the invasive margin had a significant impact on better disease-free survival. In contrast, *Salama et al. (2009)* found that patients with increased densities of CD8+ and CD45RO+ memory TILs did not exhibit any improved survival. Therefore, the prognostic value of TILs in a single region is unreliable, and more detailed insights into the spatial heterogeneity of TILs would improve the accuracy for predicting the prognosis of CRC patients.

Adjuvant chemotherapy (ACT) has been routinely recommended for CRC patients at an advanced stage to overcome postoperative recurrence, but there is no strong evidence that stage II CRC patients will achieve the same benefit from ACT (*Kannarkatt et al., 2017*; *Baxter et al., 2022*). Given the fact that approximately 10–30% of CRC patients with early stage diseases still developed local recurrence after surgical resection, novel CRC stratification frameworks are need for identifying the high-risk characteristics to aid in the clinical decision making process (*Galon et al., 2013*; *Fridman et al., 2012*). Recently, work by *Qaderi et al. (2021)* showed that ageing, pT4 tumor size and poor differentiation were the risk factors for recurrence in stage I–II CRC. However, no similar results were observed in our study. By contrast, we found that CD3+/CD8+ TILs ratio in IM region was higher in recurrence patients, whereas no significantly difference was observed in densities of CD3+

and CD8+ TILs between recurrence and non-recurrence patients. As described above, CD3+ TILs in invasive margin were mainly composed by CD4+ TILs. This observation suggested that peritumoural CD4+ TILs, rather than CD8+ TILs, had an important impact on the recurrence of CRC, in consistent with the commonly held belief that an abundance of TILs contact with tumor cells, but they often fail to control disease, partly due to the increased ratio of suppressive immune cells, such as CD4+Treg (*Hariyanto, Permata & Gondhowiardjo, 2022*). In agreement with our results, *Salama et al. (2009)* found in a large stage II patient cohort that the high density of CD4+Treg showed an opposite association with cancer-specific survival. These observations may provide a predictive indicator to identify early stage CRC patients who will benefit from ACT.

One of the limitations of this study was that it was a single-center and retrospective study that lacked a well-structured external validation. Additionally, the number of patients was not large enough, especially those with early stage of diseases. Therefore, a multicenter study with a large cohort of early stage CRC patients is required to verify these findings in the present work.

## CONCLUSIONS

In conclusion, we found that CRC patients had a more favorable PFS in cases with high densities of intratumoral and peritumoral TILs. High ratio of CD3+/CD8+ TILs in invasive margin was identified as a risk factor associated with recurrence of early- stage patients, which may help for stratifying patients and guiding personalized treatment.

## ACKNOWLEDGEMENTS

The authors thank the participants in this study for their contribution to this research.

### Funding
This study was supported by the Shenzhen Foundation of Science and Technology (JCYJ20210324111013036 and JCYJ20190809160001751), and Funds for Scholar of Shenzhen Baoan People's Hospital (BAYYZDXK001). The funders had no role in study design, data collection and analysis, decision to publish, or preparation of the manuscript.

### Grant Disclosures
The following grant information was disclosed by the authors:
Shenzhen Foundation of Science and Technology: JCYJ20210324111013036, JCYJ20190809160001751.
Funds for Scholar of Shenzhen Baoan People's Hospital: BAYYZDXK001.

### Competing Interests
The authors declare there are no competing interests.

## Author Contributions

- Lu Liu conceived and designed the experiments, authored or reviewed drafts of the article, and approved the final draft.
- Meiling Long performed the experiments, prepared figures and/or tables, and approved the final draft.
- Shengyuan Su analyzed the data, authored or reviewed drafts of the article, and approved the final draft.
- Lijun Wang analyzed the data, authored or reviewed drafts of the article, and approved the final draft.
- Jintao Liu conceived and designed the experiments, authored or reviewed drafts of the article, and approved the final draft.

## Human Ethics

The following information was supplied relating to ethical approvals (i.e., approving body and any reference numbers):

Ethical, legal, and social implications were approved by an ethical review board of People's Hospital of Baoan District

## Data Availability

The patient data is available in the Supplementary File.

## Supplemental Information

Supplemental information for this article can be found online at http://dx.doi.org/10.7717/peerj.16747#supplemental-information.

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
