# Peer review of "Clinical impact of heterogeneously distributed tumor-infiltrating lymphocytes on the prognosis of colorectal cancer"

_PeerJ, doi:10.7717/peerj.16747_

## Round 0.1 · original submission · Minor Revisions

The paper effectively demonstrates the clinical impact of TILs in CRC and their potential as prognostic indicators. The findings could have significant implications for the personalized treatment of CRC, contributing valuable insights to the field of oncology.

Kindly revise your manuscript for proper referencing and ensure that all data mentioned are properly cited.

All reviewers' comments attached should be addressed.

**Language Note:** The review process has identified that the English language must be improved. PeerJ can provide language editing services - please contact us at [email protected] for pricing (be sure to provide your manuscript number and title). Alternatively, you should make your own arrangements to improve the language quality and provide details in your response letter. – PeerJ Staff

Reviewer 1 ·

Basic reporting

1. In the table 4 the last row mentions “CD3/+/CD8+ TILsIM” the authors should correct it.
2. In the Figure 1 (I) the authors have mentioned the X axis as “CT” where as in the figure legends they have mentioned TC the authors should stick to one format.
3. Similarly, for the figure legend 1 the authors have mentioned (a-f) whereas else in the manuscript the upper case has been used the authors should stick to one format.
4. For the figures 1 G,H,I and Figure 2 A,B,C,D if the groups are showing significant difference the authors should show this by using a (*) mark between the two groups.

Experimental design

1. The authors should clarify the way of IHC (such as incubation time, secondary antibody, etc.)
2. Lymphocyte quantification in tumor sections is done manually, by two individuals (subjective measurements). More rigorous immunopathological analyses can be done automatically, by which different tumor compartments are segmented, and the total number of lymphocytes can be counted.

Validity of the findings

1. Most of the patients are lymph node-negative in the present study (57.7%). In spite of this, the recurrent rate is relatively high (48.7%). How do you interpret this?
2. The data about tumor stroma regions are not included. Even when the results do not give statistically significant values, it is appropriate to show all counting and analyses from the tumor stroma regions.
3. In the discussion section the authors should mention any limitations which are associated with the study

Additional comments

The manuscript is a pleasant read. For any solid tumor, the spatial heterogeneity is always an interesting topic. Whether TILs offered a prognostic marker for CRC had been extensively studied, but marked spatial heterogeneity has resulted in some confusion regarding their impact on the prognosis. The work done by Liu et.al provided strong support for prognostic potential of TILs in CRC. Notably, they got some new insights, the predictive value of TILs in identifying early-stage CRC patients at high risk of recurrence. Therefore, this article can be considered for publication.
Additionally, there are some issues that are worth discussing in the future
1. It will be interesting to see if the authors can draw the same conclusions using publicly available datasets such as transcriptomics.
2. I will be very happy to know the spatial heterogeneous distribution of other lymphocytes or inflammatory cells, such as B cells, macrophage, NK and DCs as well as their correlation with survival using the same patient cohort studied here.

Reviewer 2 ·

Basic reporting

1. In the material and methods section the authors have written “None patient was treated with preoperative chemotherapy or radiotherapy.” which should be None of the patients
2. The authors should label the p-value on figure 1 g-h, if any.
3. The authors should label the results of log-rank test on Figure 2.

Experimental design

1. In the Immunohistochemistry section, the authors should mention the H&E staining that often contributes to categorize tumor zones and borders.
2. Under the Cell Quantification section, the authors have mentioned they used the fields having most abundant infiltrating T cells area. Will this approach have a more biased results than using a unified field?

Validity of the findings

1. The authors should describe the differences between the tumor stroma (TS) and tumor center (TC) in detail.
2. The authors have mentioned in the introduction section that “very diverse from patient to patient” however there is growing evidence suggesting role of different subsets of T cells in CRC why do the authors see the role of only CD3+ and CD8+ cells alone? For example, in the article PMID: 35655158, it has been shown that CD4+ T cells regulate the anti-tumor immunity in CRC and that different subsets of Tregs which change the tumor micro-environment, and have different impact on DFS
3. The authors mentioned that “comprised of both CD8+ and CD4+ cells, whereas CD4+ cells predominated over CD8+ cells in invasive margin.” do the authors have not shown any data involving the CD4+ Tcells neither have they cited any article regarding this.
4. At the line “As described above, CD3+ TILs in invasive margin were mainly composed by CD4+.” the authors have not shown any data or reference for CD4+
5. In the discussion section the authors have mentioned “indirect action of secreted soluble mediators, such as cytokines, chemokines and other bioactive factors, rather than through direct T cell-mediated cytotoxicity.” they should cite the article PMID: 28555670 which reviews the roles of chemokines in cancer immunity and tumorigenesis.

Additional comments

The manuscript is well written with a brief introduction to the topic and the authors have cited the recent published studies for their purpose. The use of tables has added more information to the manuscript. The material methods section has all the necessary information. The authors have used the appropriate statistical tests. The discussion is well written.

Reviewer 3 ·

Basic reporting

no comment

Experimental design

no comment

Validity of the findings

no comment

Additional comments

This manuscript titled "Clinical impact of heterogeneously distributed tumor-infiltrating lymphocytes on the prognosis of colorectal cancer" focuses on the role of tumor-infiltrating lymphocytes (TILs) in colorectal cancer (CRC), in which the authors conducted a clinical retrospective study to analyze the distribution of CD3+ and CD8+ TILs in CRC tissues and their association with clinical outcomes. Overall, this study addresses an important aspect of CRC prognosis by focusing on the immune response within the tumor microenvironment. The methodology is relatively robust, involving detailed immunohistochemical analysis and statistical evaluation. The paper effectively demonstrates the clinical impact of TILs in CRC and their potential as prognostic indicators. The findings could have significant implications for the personalized treatment of CRC, contributing valuable insights to the field of oncology.
Here are some of my comments listed as follows. I hope the author can consider and adopt to improve the quality of this manuscript.
1. Line 181 and Figure 1G and 1H: That's a bit of a puzzling statement. The current display (Figure 1G and 1H) can be misleadingly thought of as a comparison of densities among IM, TC, and TS groups. If the purpose is to show the correlation between IM and TC, then it would be better to use a scatter plot with the axes as the two variables.
2. In the results section, the author focuses too much on describing the observed data rather than elaborating on the interconnections of the results. A certain amount of summary in each section would have helped the reader to deepen their knowledge and understanding of the results.
3. I would suggest that the authors mark groups with statistically significant differences with an asterisk (*) or a p-value in Figures 1 and 2, which can make it easier for the reader to understand.

---

## Round 0.2 · accepted · Accept

Thanks for revising your manuscript according to the reviwers comments which were all adressed. I am pleased to inform you that your manuscript is now ready for publication.

Reviewer 1 ·

Basic reporting

no comment

Experimental design

no comment

Validity of the findings

no comment

Additional comments

Compared to the first submission, the authors have modified figures, provided a much more detailed description of the Immunohistochemistry and added the limitation discussion on their study.
The association between TILs in the tumor stroma and clinical features were shown in the supplementary table which allows having a better overview.
The spatial heterogeneity of immune cells and their potentially predictive and prognostic power are hot topics. I think the authors could have gone further in the clinical aspect.

Reviewer 2 ·

Basic reporting

NO

Experimental design

NO

Validity of the findings

NO

Additional comments

The authors had modified the manuscript as suggested. A new paragraph regarding the work limitation was added in the discussion. The revision of results and discussion made the paper more readable. Therefore, at present form the paper is suitable for publication.